# Recent Updates on ALMT Transporters’ Physiology, Regulation, and Molecular Evolution in Plants

**DOI:** 10.3390/plants12173167

**Published:** 2023-09-04

**Authors:** Siarhei A. Dabravolski, Stanislav V. Isayenkov

**Affiliations:** 1Department of Biotechnology Engineering, Braude Academic College of Engineering, Snunit 51, Karmiel 2161002, Israel; sergedobrowolski@gmail.com; 2Institute of Agricultural and Nutritional Sciences, Martin Luther University Halle-Wittenberg, Betty-Heimann-Strasse 3, 06120 Halle, Germany; 3Department of Plant Food Products and Biofortification, Institute of Food Biotechnology and Genomics, The National Academy of Sciences of Ukraine, Osipovskogo Str. 2a, 04123 Kyiv, Ukraine

**Keywords:** ALMT, anion channel, aluminium tolerance, malate transport, STOP1, phosphate deficiency

## Abstract

Aluminium toxicity and phosphorus deficiency in soils are the main interconnected problems of modern agriculture. The aluminium-activated malate transporters (ALMTs) comprise a membrane protein family that demonstrates various physiological functions in plants, such as tolerance to environmental Al^3+^ and the regulation of stomatal movement. Over the past few decades, the regulation of ALMT family proteins has been intensively studied. In this review, we summarise the current knowledge about this transporter family and assess their involvement in diverse physiological processes and comprehensive regulatory mechanisms. Furthermore, we have conducted a thorough bioinformatic analysis to decipher the functional importance of conserved residues, structural components, and domains. Our phylogenetic analysis has also provided new insights into the molecular evolution of ALMT family proteins, expanding their scope beyond the plant kingdom. Lastly, we have formulated several outstanding questions and research directions to further enhance our understanding of the fundamental role of ALMT proteins and to assess their physiological functions.

## 1. Introduction

Aluminium-activated malate transporter (ALMT) proteins play vital roles in plant adaptation to aluminium toxicity and acid soils. Soil acidification is characterised by the increased mobility of Al^3+^ ions, which tend to bind phosphorus into stable complexes, thus combining Al^3+^ toxicity and decreased phosphate bioavailability. Therefore, the release of malate allows for neutralisation of Al^3+^ and the disengagement of phosphates from the complex. Early experiments indicated that AtALMT1 is regulated by low pH, H_2_O_2_, auxin, and abscisic acid (ABA) [1]; Ca^2+^ [2]; and γ-Aminobutyric acid (GABA) [3]. Further experiments with soybean ALMT1 demonstrated that the presence of adequate phosphate levels is crucial for soybean adaptation to acid soils [4]. On the contrary, ethylene was shown to act as a negative regulator of the Al^3+^-induced malate efflux [5].

Recently, the expression of the *Arabidopsis thaliana* ((L.) Heynh). *ALMT* gene family under phosphorus-deficient conditions was characterised. In contrast to early experiments, only *AtALMT3* was significantly up-regulated in the root by low-phosphate conditions. Specifically, the *AtALMT3* gene was expressed in the root hair cells and protein was localised in the plasma membrane and small vesicles. *AtALMT3* knock-down and overexpressing lines demonstrated reduced malate exudation and increased malate exudation, respectively, under phosphate-deficient conditions. Therefore, these results indicate the crucial role of plasma membrane-localised *AtALMT3* in low-phosphate-inducible malate exudation in Arabidopsis root hair [6]. Interestingly, the ALMT1 homologue from *Lupinus albus* L., which was expressed in the root apices under phosphorus deficiency, was moderately repressed by Al, involved in Al-independent malate release from root tips, and participated in cation/metal root-to-shoot translocation [7].

The regulatory role of abscisic acid under Al^3+^ stress was recently updated. Experiments on rice bean (*Vigna umbellata* (Thunb.) Ohwi and H. Ohashi) and Arabidopsis *abi5* (ABA-Insensitive 5) mutants demonstrated that ABA-mediated Al stress response is not associated with known Al tolerance mechanisms (AtALMT1 and MULTI-DRUG AND TOXIC COMPOUND EXTRUSION (AtMATE)). Interestingly, nearly 1/3 of genes activated by Al stress and ABA treatment are the same. Furthermore, ABI5 was identified as a transcription factor that mediates Al stress tolerance via regulation of these genes, most importantly, osmoregulation and cell wall modification-related genes (such as *Peroxidase 72*, *Stress*-*associated protein 12*, *XYLOGLUCAN ENDOTRANSGLUCOSYLASE*/*HYDROLASE 23*, and *GLUTATHIONE S-TRANSFERASE TAU 8*), suggesting the presence of another ABA-dependent mechanism of Al tolerance in plants [8].

Additionally, experimental results obtained from *A. thaliana* and *Amaranthus mangostanus* (L.) suggested that the expression of *ALMT* genes is up-regulated by salt stress, thus increasing the release of malate to the rhizosphere [9,10]. Similarly, *Guzmania monostachia* (L.) Rusby ex Mez) plants supplied with ammonium and exposed to drought stress showed higher expression of ALMT1, increased accumulation of soluble sugars involved in osmotic adjustment, and higher antioxidant enzyme activity (SOD, CAT, APX, and GR) [11]. Also, the comparison of Al^3+^-tolerant and Al^3+^-sensitive wheat (*Triticum aestivum* L.) lines demonstrated that roots of tolerant lines exuded more malate and GABA under high pH and Al treatment, thus acidifying the rhizosphere faster. While the expression of TaALMT1 did not change under high pH, the expression of GABA synthesis genes (TaGAD and TaGAD1) was increased and the expression of the GABA catabolism gene (GABA-T) was decreased in the tolerant line [12].

Thus, the physiological roles of ALMT transporters are substantially wider than was thought before. This type of anion transporter might play a crucial role in physiological adjustments during adaptation to a wide range of stresses. However, many sides of their functionality, physiology, regulation, and interaction remain largely unknown. In this review, we summarise the recent knowledge on the ALMT family and discuss molecular mechanisms regulating their functioning and involvement in various physiological processes. Finally, we conducted a rigorous bioinformatic and molecular evolution analysis of the ALMT protein family using currently available sequencing data, solved crystal structures, and bioinformatic tools.

## 2. Bioinformatic Analysis of ALMT Family Proteins

The common plant model organism *A. thaliana* has 14 *ALMT* genes which encode 16 proteins (*ALMT14* has three isoforms). Examination of ALMTs at the amino acid level suggests the presence of several domains. The aluminium-activated malate transporter (pfam11744) domain (app 400aa) itself is the major component of the ALMT proteins. Further, the ALMT domain could be divided into the N-terminal pore-forming part (transmembrane domain (TMD)), which contains six transmembrane helices, and the C-terminal cytosolic domain (CTD) (Figure 1). ALMT11 has only a partial ALMT domain, and, despite it having well-conserved TM1 and TM2, its ability to transport malate or other substrates is still neither confirmed nor disproved. Therefore, *ALMT11* is usually considered as a truncated gene which produces non-functional protein. Also, several ALMT proteins have a deletion in the CTD (in particular, ALMT3-6, 9, 12, and 13 and all isoforms of ALMT14). To the best of our knowledge, the functional significance of this deletion is still unknown (Appendix A).

Additionally, another domain has been identified in AtALMTs with a high level of confidence (*p*-value 0.005 and higher) by NCBI conserved domain database search tool [13]. All ALMT proteins (except ALMT14 isoforms 2 and 3) have a fusaric acid resistance protein-like (pfam13515) domain, which ranges from TM2 to TM6, thus basically corresponding to the pore-forming part (Figure 1). Interestingly, the ALMT14-2 isoform almost completely lacks TM3, while ALMT14-3 lacks TM3 and partially lacks TM4. To the best of our knowledge, the functional properties of these isoforms have never been tested experimentally. Considering that TM3 and TM4 contain the important conservative residues necessary for malate recognition and transport, the functional role of ALMT14 isoforms 2 and 3 would be very interesting to investigate.

### 2.1. Malate and Al Recognition and Transport

Recently, the structures of Arabidopsis ALMT1 and *Glycine max* ((L.) Merr.) ALMT12 have been characterised. Also, residues participating in the recognition and transport of malate and Al were identified [14,15]. Besides residues involved in Al binding and coordination (Appendix A) located on the TMD, several negatively charged Al-responsive residues were earlier identified on the CTD (Appendix A). Interestingly, glutamate 276 (E276 for AtALMT12) was found to be crucial for channel activity, while in TaALMT1 and AtALMT1 (E284 and E256, respectively) this residue was spotted only as required for Al-induced malate transport [16,17].

Based on the conducted analysis (Appendix A), residues involved in malate recognition are strongly conserved, while for Al-binding and Al-coordinating residues there were some discrepancies. Thus, conserved D59 (for AtALMT1) could be found only in ALMT1, 2, 7, and10; E156 was only found in ALMT1 and 7; D160 was only found in ALMT1 and 2; and Al coordination M60 was only found in ALMT1, 2, 7, and 10–14 (all isoforms of ALMT14). Additionally, several conservative sites have been identified in GmALMT12, which formed a positively charged ring at the pore entrance (R187 and R198) and a second positively charged ring within the membrane (K109, R113, and R158), mediated an interhelical network within the pore (E100, D168, and Y169), and stabilised the bottom of the T-shaped pore (K164 and K165). Also, the conserved W90 was found to act as a switcher, regulating channel gating (Appendix A) [15].

### 2.2. ALMT/GABA Interplay

The role of GABA as an ALMT regulator and the role of ALMTs in GABA binding and transport has been studied in detail in recent years. The identified conserved GABA-binding motif [3] was shown to exhibit minor variation, which probably greatly affects the interaction of individual ALMTs with GABA. Thus, mutation of the first and second aromatic residues (F213 and F215 for TaALMT1) reduced interaction with and transport of GABA, while preserving positive activation by Al^3+^ and by external anions [3,18]. Interestingly, in Arabidopsis, only ALMT5 and 6 have two aromatic amino acids in these positions (both F) in addition to ALMT9 (F and Y). However, so far, no studies have confirmed the functional interchangeability of F and Y amino acids. Also, recent research suggests that AtALMT9 does not interact with GABA directly [19]. Furthermore, the significance of these and other sites in the GABA recognition motif involving ALMTs should be specified (for example, FFC in ALMT10 or LFF in ALMT12).

### 2.3. CaM Binding by ALMTs

Recent experiments on ALMT12 from *Brachypodium distachyon*, the closest transporter homologous to *Arabidopsis thaliana* ALMT12, have identified specific residues in the CTD which are responsible for binding with calmodulin (CaM). Thus, substitutions of R335A, R338A, and K342A greatly decreased the ability of malate to activate the channel and binding to CaM. Comparison of different substitution variants (triple and double) helped to emphasise the dominant role of the R335 site [20]. Interestingly, none of the Arabidopsis ALMTs has all three (R335, R338, and K342) conservative sites, while residues corresponding to R335 and K342 are rather well matched (Appendix A).

### 2.4. ALMT Phosphorylation

Earlier research tested several potential protein kinase C (PKC)-mediated phosphorylation sites on TaALMT1 (S56, S183, Y184, T323, S324, S337, S351-352, and S384). However, only the S384A mutation resulted in a noticeable change in transport properties [21]. Recently, the role of ALMTs in abiotic stress response was demonstrated in experiments on plants subjected to drought stress and ABA treatments. Interestingly, AtALMT4 mediated malate efflux during the drought and ABA-induced stomatal closure depended on the phosphorylation of S382 [22]. This site corresponds to S384 in TaALMT1 and is rather conserved in Arabidopsis (except for ALMT9, ALMT6, and ALMT3) (Appendix A). However, the significance of phosphorylation at this particular position for the regulation of other Arabidopsis ALMTs was not shown and remains a question for future experiments.

## 3. Molecular Evolution

### 3.1. Sequence Identification

To track the evolutionary origin of the ALMT, we have searched for homologous sequences in the Uniprot [23] (http://www.uniprot.org/ accessed on 25 April 2023), Pfam/InterPro [24,25] (http://www.ebi.ac.uk/interpro accessed on 25 April 2023) and NCBI databases [26] (http://www.ncbi.nlm.nih.gov/ accessed on 25 April 2023) using both the amino acid-based BLAST and the domain-based search methods [27]. During the sequence search, the truncated, partial, and identical sequences were removed. In total, 4650 sequences were identified (*Viridiplantae*: 4646; *Amoebozoa*: 3; *Opisthokonta*: 1). Many bacterial and archaea proteins have been annotated as containing an ALMT domain; however, closer examination disproved the presence of a full-length ALMT domain. Also, while AtALMT11 has no full-length ALMT domain, it was not used in the tree reconstruction.

### 3.2. Phylogenetic Analysis

To understand the evolutionary history of the ALMT proteins, we reconstruct the phylogenetic trees with the maximum likelihood method. *Opisthokonta* and *Amoebozoa* ALMT proteins were used as an out-group, and 240 ALMT proteins from different taxa were selected from the *Viridiplantae* group (Appendix A). Because we used only sequences from full-length ALMT proteins, AtALMT11 was not used for tree reconstruction. In general, we can define eight separated clusters and the number of sub-clusters of closely related proteins (Appendix A). Therefore, ALMTs from amoeba and *Sphaeroforma arctica* were located on a separate branch and counted as clade I. Clade II was comprised algae, mosses, and ferns (*Chara braunii*, *Klebsormidium nitens*, *Physcomitrium patens*, *Ceratodon purpureus*, and *Marchantia polymorpha*), while sub-clade IIa included only the fern *Ceratopteris richardii*. Clade III was well separated and formed by ALMT9-like proteins from plants of different taxa (including also both *Eudicotyledons* and *Monocotyledons*, such as *Oryza species*, *Phaseolus vulgaris*, and *Cucurbita maxima*) (Appendix A).

Clade IV had three well-separated sub-clades, with IV formed by ALMT proteins from Eudicotyledon species (including *A. thaliana* ALMTs 12, 13, and 14), sub-clade IVa comprising only proteins from *Monocotyledon* species (such as *Oryza* sp., *Triticum aestivum*, *Hordeum vulgare* and others), and sub-clade IVb including ALMTs from several *Eudicotyledon* species and species from other taxa (*Nicotiana tabacum*, *Solanum lycopersicum*, *Nelumbo nucifera*, *Cinnamomum micranthum*, and others). Similarly, clade V was formed mostly by ALMTs from *Eudicotyledon* species (including *A. thaliana* ALMT10) and proteins from other taxa (*Aquilegia coerulea*, *Thalictrum thalictroides*, and *Nelumbo nucifera*). Clade Va contained ALMTs from *Monocotyledon* species, with ALMTs from *Cinnamomum micranthum* out-grouping the sub-clade consisting of *Amborella trichopoda* species, i.e., the entire Vth clade (Appendix A).

Clades VI and VIa were composed of proteins from *Eudicotyledon* species, including AtALMTs (ALMT1, 2, and 7 for clade VI and ALMT8 for VIa). Sub-clade VIb was formed mostly by proteins from the *Monocotyledon* species, and ALMTs from *Kalanchoe fedtschenkoi* and *Cinnamomum micranthum* acted as out-groups for this sub-clade. The small sub-clade Vic contained ALMT proteins from different species (*Thalictrum thalictroides*, *Nelumbo nucifera*, and *Tetracentron sinense*). Clade VII had no sub-clade and contained ALMT proteins from *Eudicotyledon* species and species of other taxa (*Prunus avium*, *Glycine max*, *Nelumbo nucifera*, *Cinnamomum micranthum*, and others). The VIIIth clade had several well-separated sub-clades, with sub-clades VIII, VIIIa, and VIIIb formed only by ALMTs from *Eudicotyledon* species (including AtALMT4-6 belonging to VIII), while VIIIc contained only *Monocotyledon* species. Furthermore, VIIId included AtALMT3 and 9 and various ALMT proteins from different taxa (*Eudicotyledons*, *Monocotyledons*, and others). Finally, sub-clade VIIIe was formed by ALMT proteins from *Aquilegia coerulea*, *Thalictrum thalictroides*, *Wollemia nobilis*, and *Araucaria cunninghamii*. Also, some ALMT proteins (such as *Amborella trichopoda*, *Selaginella moellendorffii*, *Marchantia polymorpha*, and *Ceratopteris richardii*) were not assigned to any clade/sub-clade and acted as an out-group for corresponding clades and sub-clades (Appendix A).

In total, our results confirmed and further expanded the phylogenetic tree published earlier [28,29], where only three “higher plant” species were used for tree reconstruction and *Arabidopsis* ALMTs were clustered into four groups: (1) ALMT8, 1, 2, and 7; (2) ALMT10; (3) ALMT12, 13, and 14; and (4) ALMT 9 and 3–6 [28]. Thus, our results further expanded these data, separating ALMT8 from ALMT1, 2, and 7 and separating ALMT3 and 9 from ALMT4-6 into different sub-clades (Appendix A).

Here, we have to disprove some earlier reports [28,30] that propose that ALMTs are a plant-specific family. We found a full-length ALMT domain in one protein from *Sphaeroforma arctica* (*Opisthokonta*) and in three proteins from Amoeba *Dictyostelium discoideum* (*Amoebozoa*). Despite obvious differences because of the distant evolutional relation, most conservative sites are presented (Appendix A). However, the role of these proteins in malate/GABA/Al^3+^ transport should be confirmed experimentally. Also, we disprove earlier reports linking ALMTs to the Aromatic Acid Exporter (ArAE) family because we found the ArAE domain only in AtALMT9 and 12 and a limited number of other species (such as A0A2R6WEU1 from *Marchantia polymorpha* and A0A1Y1HXE2 from *Klebsormidium nitens*). On the contrary, we can define ALMTs as part of a large group related to the fusaric acid resistance proteins (pfam13515) (Appendix A). This group is widely presented in bacteria and fungi, and, considering the presence in other proteins with a fusaric acid resistance protein-like domain and a fusaric acid resistance protein family (PF04632) domain, such as in *Arabidopsis* and other species (importantly, without an ALMT domain), it is more plausible that ALMTs are part of or evolutionary originated from this group. Also, clear functional separation of the ALMT domain on the N-terminal pore-forming part (thus, acquired from fusaric acid resistance proteins) and the C-terminal CTD (which was, most probably, acquired later from another source and which performs a more pronounced regulatory role) further corroborates this hypothesis. However, this question requires further detailed investigation, which is far beyond the topic of this review.

In total, the results of our phylogenetic analysis confirmed the general tree topology reported earlier. At the same time, the position of individual ALMTs has been assigned to separate sub-clades. Because we have identified several non-plant ALMT proteins and the presence of the fusaric acid resistance domain in all ALMT proteins, we have disproved that ALMTs are a plant-specific protein family related to ArAE that possesses an ArAE domain. Therefore, the evolutionary origin of the ALMT family proteins requires further detailed investigation.

## 4. ALMTs in the Regulation of Stomata/Guard Cells

Anion channels localised on the plasma membrane of guard cells are the key contributors to stomata movement through the release of anions and subsequent membrane depolarisation [31]. Early research has demonstrated that two members of the ALMT family are localised to the tonoplast (AtALMT6 and AtALMT9, but not exclusively) and plasma membrane (AtALMT12) of the guard cells, thus suggesting the role of these channels in vacuolar malate transport or across the plasma membrane, respectively (Appendix A) [32,33,34]. Accordingly, AtALMT6 was characterised as a Ca^2+^-activated transporter, aiming a malate flux from the cytosol to the vacuole. However, *Atalmt6* plants had no distinctive phenotype, possibly because of the functional redundancy of other ALMT-family transporters in guard cells [33]. AtALMT9 was defined as a Cl^−^ channel, which is activated by cytosolic malate and involved in regulating stomata opening, albeit not closing. Interestingly, *Atalmt9* plants showed a drought-resistant phenotype, which is congruent with impaired stomatal opening due to decreased uptake of Cl^−^ in guard cells [35]. AtALMT12 was characterised as a quick anion channel, which possesses a high capacity for the transport of Cl^−^, NO_3_^−^, and malate from the guard cells and exhibits negligible reaction on extracellular Al^3+^. Furthermore, the *Atalmt12* mutant inertly closes stomata in response to CO_2_, ABA, darkness, dehydration, or Ca^2+^. Thus, these data suggest the involvement of AtALMT12 in regulating stomata closure rather than in Al^3+^ resistance [32,36]. 

Recently, the *Atalmt6* mutant phenotype was described with Cl^−^-dependent defective stomatal opening induced by blue light and fusicoccin, a fungal diterpenoid glycoside with a strong ability to activate plasma membrane H^+^-ATPase [37]. Furthermore, nucleotides were identified as effective modulators of AtALMT9 transport activity. Interestingly, ATP was the most effective cellular blocker and ATP hydrolysis was not required, suggesting that anion transport across the vacuolar membrane is regulated by cytosolic nucleotides and the energetic status of the cell [38]. Furthermore, under NaCl stress, *Atalmt9* mutants showed reduced shoot accumulation of both Cl^−^ and Na^+^, while plants complemented with the E196A channel variant (point mutation that exhibits increased Cl^−^ current activity [39]) showed enhanced channel activity and higher Cl^−^ and Na^+^ accumulation. Considering that NaCl up-regulated *ALMT9* expression in the vasculature of shoots and roots, this suggests it has a crucial role in regulating the entire vacuolar Cl^−^-loading of the plant immediately after salt-stress is encountered [40]. The *Atalmt12* plants responded with slower stomatal closure with a high CO_2_ concentration and in darkness but also accumulated higher levels of malate and fumarate and had improved mesophyll conductance. Interestingly, these effects were associated with increased photosynthesis and respiration rates, which resulted in improved growth [41]. Also, recent experiments on ALMT12 from the grass *Brachypodium distachyon* ((L.) P.Beauv.) (BdALMT12) demonstrated that both Ca^2+^/calmodulin and malate are co-regulators and required for channel activation (Figure 2) [20].

Plant stress hormone ABA is known to regulate plant drought stress adaptation through various pathways. In guard cells, ABA regulates AtALMT12 via kinase OPEN STOMATA 1 (OST1) and protein phosphatase ABSCISIC ACID-INSENSITIVE-1 (ABI1) [42]. Later experiments have defined that xylem-delivered sulphate could trigger drought-mediated stomata closure in several pathways (Figure 2). First, sulphate can reach guard cells and activate AtALMT12 directly. Second, sulphate in guard cells can induce the expression of *9-cis-epoxycarotenoid dioxygenase* (*NCED3*), the key step in chloroplastic ABA biosynthesis. Third, sulphate can stimulate ABA biosynthesis through cysteine synthesis, thus promoting the activity of abscisic aldehyde oxidase 3 (AAO3)—the last step in ABA biosynthesis [43]. Furthermore, ABA regulates AtALMT4, another member of the ALMT family mediating Mal^2-^ release from the vacuole in an ABA-dependent way. *Atalmt4* plants demonstrated impaired stomatal closure in response to ABA and increased wilting in response to drought stress and ABA. Interestingly, the activity of AtALMT4 depended on phosphorylation on the C-terminal S382, which can be phosphorylated by mitogen-activated protein kinases in vitro. Dephosphomimetic mutants of AtALMT4 S382 showed increased channel activity and Mal^2-^ efflux, while phosphomimetic mutants were electrically inactive and phenocopied the *almt4* mutants [22].

Because ALMT proteins have a GABA-binding motif [3], it was suggested that GABA can directly regulate ALMT activity. Experiments on plant protoplasts and monkey kidney cells confirmed that GABA treatment resulted in AtALMT12 current reduction [44]. Interestingly, experiments on *Triticum aestivum* (L.) root tips and *TaALMTs*-expressing *Xenopus laevis* oocytes demonstrated that ALMTs can transport both GABA and anions *in planta*, as well as that an intact GABA-binding motif is crucial for its normal functioning [18]. Similarly, GABA was shown to reduce the anion channel opening frequency, thus regulating anions passing through the pore under stress [45]. Further research suggested that GABA can affect stomata aperture (thus regulating water use efficiency and drought resilience) by modulating the activity of ALMT family proteins [46]. However, it is most likely that GABA acted on ALMTs not directly, but through other unknown regulatory factors or vacuolar ion channels/transporters (Figure 2) [19]. This point of view is supported by the substantial R-type anion currents in the double (*Atalmt12/13* and *almt12/14*) and triple (*Atalmt12/13/14*) mutants [47]. Indeed, our data analysis of co-localised, co-expressed, and domain-sharing proteins/genes of ALMTs suggested 27 various transporters which could potentially be involved in regulating stomata aperture. In this list, we can find some well-known transporters (like members of the ABC family, SLAH, SKOR, SWEET), as well as several unknown and uncharacterised transporters (Appendix A). Thus, further study of the interaction of ALMTs with other membrane transport proteins will perhaps shed light on the coordinated mechanism of transmembrane ion movement during stomata movement.

Therefore, the current model suggests the key role of vacuolar ALMTs during stomatal opening. When K^+^ enters the cells, starch is degraded to malate for the further compensation of charges [28]. As a result of starch degradation, the elevated cytosolic concentration of malate leads to the activation of ALMT9 combined with subsequent intensification of malate transport into the vacuole. In addition, ALMT9 is also activated to mediate the transport of Cl^−^ into the vacuole and cause an accumulation of solutes in the vacuole of guard cells, with the final result being stomatal opening [28,35]. Furthermore, existing experimental data support the point of view that, as concerns the vacuolar chloride channel CLCa, the activity of AtALMT9 is crucial in stomata opening through the transport of Cl^−^ into the vacuole [35,40,48,49].

## 5. Transcriptional Regulation

Regulation of the transcription activity of ALMT-encoded genes is the most crucial process in the physiological and biochemical adjustment of plants during growth and development and stress adaptation. One of the central roles in the transcriptional regulation of ALMT genes has been appointed to transcription factors (TFs) such as STOP1 (SENSITIVE TO PROTON RHIZOTOXICITY) (Figure 3). STOP1 belongs to C2H2-type TFs, and it was identified as a crucial regulator of Al-induced expression of the *ALMT1* gene in Arabidopsis and other plant species [50,51]. Accordingly, the WRKY46 transcription factor was identified as a repressor of *AtALMT1*. *Wrky46* plants demonstrated higher ALMT1 expression and increased malate exudation, with a subsequently higher tolerance to Al [52]. In the following years, the mechanism regulating transcriptional activation/repression of *ALMT* was further elucidated.

### 5.1. STOP1/CAMTA2/WRKY Regulatory Node

In planta complementation assays of AtALMT1 demonstrated that thepromoter region between −540 and the ATG codon contains eight *cis* elements essential for Al induction and STOP1 regulation. In particular, the region around the −297 site (*cis*-D) is recognised by four zinc finger domains of STOP1; thus, it is necessary for STOP1 binding. Some other *cis* elements have also been characterised, with *cis*-B shown to interact with an unknown repressor and *cis*-C with the activator CALMODULIN-BINDING TRANSCRIPTION ACTIVATOR 2 (CAMTA2). Accordingly, *cis*-H is crucial for the early phase, while *cis*-A and *cis*-C are important for the late phase of Al^3+^-stress response [53]. Also, calcineurin B-like protein (CBL) was shown to up-regulate *ALMT1* expression and regulate root malate efflux [54]. CALMODULIN-LIKE24 (CML24) is another protein involved in Al-induced cytosolic Ca^2+^ signal transmission to regulate *ALMT1* expression in a STOP1-independent way. Upon activation, CML24 interacted with CAMTA2 to promote ALMT1-mediated malate exudation from roots (Figure 3). Also, CML24 suppressed *WRKY46*, thus preventing its repression of *ALMT1*. Therefore, these results provide a novel player in Ca^2+^-mediated signalling in ALMT1-dependent Al stress resistance [55].

Interestingly, the application of phosphatidylinositol 4-kinase (PI4K) and phospholipase C (PLC) inhibitors suppressed *AtALMT1* transcription and reduced Al-activated malate transport. Furthermore, AZD7762, a human protein kinase inhibitor, suppressed late-phase Al-induced *AtALMT1* expression, presumably by inhibiting homologous calcineurin B-like protein (CBL)-interacting protein kinase (CIPK) and/or Ca-dependent protein kinase (CDPK). Also, AZD7762 induced the expression of *WRKY46 (ALMT1* repressor*)* and suppressed *CAMTA2 (*an *ALMT1* activator*)* (Figure 3). These data suggest that phosphatidylinositol metabolism is involved in regulating malate secretion in plants under Al stress [56].

Recent research has demonstrated that transgenic plants expressing the *WRKY21* TF from soybean (*Glycine max* (L.) Merr.), when treated with AlCl_3_, showed higher induction of Al stress-associated genes (*STOP1*, *ALMT1*, *MATE*, and *ALS3*), along with up-regulated expression of other stress-related genes (from the *COR*, *DREB*, *LEA*, and *RD* families). Also, *GmWRKY21*-expressing transgenic Arabidopsis plants had increased root growth, lower MDA, and higher proline accumulation under AlCl_3_ stress. These results suggest that other TFs may promote tolerance to Al stress through the STOP1–ALMT1 node as a part of the general abiotic stress response [57].

### 5.2. STOP1/STOP2 Pathway

Recently, analysis of GFP-tagged STOP1 proteins helped to identify other genes rapidly induced upon Al treatment and containing STOP1-binding sites, such as *STOP2* and *GLUTAMATE*-*DEHYDROGENASE1* and 2 (*GDH1* and 2). Furthermore, the *GDH1* and *GDH2* genes were suppressed in *stop1* mutant lines treated with AlCl_3_, and the *gdh1/2* double mutant showed increased sensitivity to Al stress, suggesting the involvement of both genes in Al tolerance in a STOP1-dependent way (Figure 3) [58]. Indeed, the role of *STOP2*, a STOP1 homologue, in Al stress tolerance and low pH was demonstrated in Arabidopsis. Expression of *STOP2* in *stop1* plants partially rescued growth and root tip viability by activating the expression of some genes regulated by STOP1. Thus, *polygalacturonase*-*inhibiting protein 1 and 2* (*PGIP1* and *PGIP2*) genes, which are known to stabilise cell walls at low pH, were activated, while the expression of *AtALMT1* was not recovered [59].

Recently, an MADS-box TF from soybean (GsMAS1) was shown to participate in Al stress tolerance. Accordingly, the expression of *GsMAS1* in Arabidopsis enhanced tolerance to Al stress, improved root length, and increased proline accumulation under AlCl_3_ treatment. In particular, *STOP1*, *ALMT1*, *STOP2*, *AtMATE*, and *PGIP2* were the key genes activated by *GsMAS1*, suggesting that it may increase resistance to Al stress through several pathways in Arabidopsis [60].

### 5.3. STOP1/Phosphate Interplay

The low level of external phosphate is another factor regulating the transcription of *STOP1* and its direct target *ALMT1*. Phenotypically, limited phosphate (_l_Pi) supply is manifested by inhibiting primary root growth in many plant species. Mechanically, _l_Pi enhances expression *ALMT1*, which subsequently promotes quick primary root growth arrest together with LOW PHOSPHATE ROOT (LPR) ferroxidases and peroxidase-dependent cell wall stiffening (Figure 4). Furthermore, the accumulation of Fe and callose in the stem cell niche inhibited cell proliferation while accelerating cell differentiation, which resulted in the meristem’s reduction. Importantly, malate exudation is crucial for Fe accumulation in the apoplast of meristematic cells and the subsequent peroxidase-dependent activation differentiation of meristematic cells in response to _l_Pi supply [61,62]. Later research defined that the tonoplast-localised ALUMINIUM-SENSITIVE3/SENSITIVE TO ALUMINIUM RHIZOTOXICITY1 transporter complex (ALS3/STAR1) repressed nuclear STOP1 accumulation, thus inhibiting the STOP1–ALMT1 pathway [63]. Furthermore, Al^3+^ can act similarly to Fe and inhibit proteasomal degradation of STOP1, thus facilitating its stability, accumulation in the nucleus, and *ALMT1* expression [64].

The MEDIATOR (MED) complex is a multi-protein assembly that controls gene expression through interaction with basal RNA polymerase II transcription machinery, *cis* regulatory elements, and enhancers. The MED16 subunit was shown to regulate plant development and response to multiple biotic and abiotic stresses, including Fe uptake (Figure 4) [65]. As was recently shown, MED16 is also involved in _l_Pi response through direct interaction with STOP1 and co-regulation of STOP1-regulated genes. Accordingly, the expression of *ALMT1* was greatly reduced in *med16* plants in both conditions (_l_Pi and normal Pi levels). These data add another transcriptional activator to the STOP1–ALMT1 node, further linking Fe uptake, _l_Pi, and Al stress response to malate efflux [66].

Recently, the AtWRKY33 TF was identified as a negative regulator of _l_Pi-induced root architecture remodelling (Figure 4). *Atwrky33* plants showed increased sensitivity to _l_Pi via inhibition of primary root growth and the promotion of root hair formation. In particular, AtWRKY33 negatively regulated *ALMT1* transcription (but not *STOP1*) under _l_Pi conditions, thus mediating Fe^3+^ accumulation in root tips to inhibit root growth [67]. Also, _l_Pi down-regulated BRASSINAZOLE-RESISTANT 1 (BZR1) expression in *A. thaliana*, a crucial regulator of the brassinosteroid signalling pathway [68]. As it was shown on *Nicotiana benthamiana* and *Oryza sativa*, nuclear-localised BZR1 competed with STOP1, thus suppressing ALMT1 activation and malate secretion [69].

Furthermore, _l_Pi inhibited primary root growth via post-transcriptional regulation. Particularly, under _l_Pi conditions, histone H3 acetylation was increased, and HDC1 protein abundance was reduced because of 26S proteasome-mediated degradation. At the same time, the histone deacetylase complex 1 (*hdc1*) plants were hypersensitive to Pi deficiency, with inhibited primary root growth and an increased number of root hairs (Figure 4). Also, under _l_Pi conditions, *hdc1* plants accumulated a higher amount of Fe^3+^ in the root tips and showed increased expression of *ALMT1*, *STOP1*, *PHOSPHATE TRANSPORTER1*, and *LPR1* and *2*. Additionally, _l_Pi conditions enriched the histone H3 acetylation of *ALMT1* and *LPR1*. These results suggest a new chromatin-level control mechanism via repression of the STOP1–ALMT1 node and associated genes in response to _l_Pi conditions through HDC1-mediated histone H3 deacetylation [70].

### 5.4. TFs Regulating ALMTs in Berries/Fruits

Despite the role in Al stress and _l_Pi response, malate greatly affects the palatability of the edible tissues of many agricultural species, such as apples, tomatoes, pears, kiwis, and others [71]. Several recent papers have defined different TFs regulating malate accumulation in the fruits of several species. Thus, in apple (*Malus domestica* Borkh.), MdMYB73 was found to activate MdALMT9, vacuolar ATPase subunit A (MdVHA-A), and vacuolar pyrophosphatase 1 (MdVHP1). Interestingly, *cold*-*induced bHLH1* (*MdCIbHLH1*) was also shown to interact with MdMYB73 and enhanced its activity, thus modulating vacuolar malate accumulation and pH in apple [72].

Furthermore, the member of the plant-specific BTB-TAZ DOMAIN PROTEIN 2 (BT2), which contains an N-terminal bric-a-brac/tramtrack/broad complex (BTB), a transcriptional adapter zinc finger (TAZ) domain, and a C-terminal calmodulin-binding (CaMB) domain, was shown to regulate malate accumulation and vacuolar pH in response to nitrate. Thus, in response to nitrate treatment, MdBT2 directly interacted with and ubiquitinated MdCIbHLH1 and MdMYB73 through the ubiquitin/26S proteasome pathway in both in vitro and in vivo conditions. Subsequently, degradation of the MdCIbHLH1 and MdMYB73 proteins reduced the levels of downstream genes, such as *MdVHA-A*, *MdVHP1*, and *MdALMT9*. These results provided the direct link between high nitrate supply and malate accumulation and vacuolar acidification in apples [73,74].

### 5.5. Regulation of STOP1 Proteasomal Degradation

STOP1 Ubiquitination. Interestingly, STOP1 regulated its degradation by up-regulating the *RAE1* gene (*regulation of Atalmt1 expression 1*), which ubiquitinated and degraded STOP1, thus forming a negative feedback loop between STOP1 and RAE1 (Figure 4) [75]. Later, similar biological functions were assigned to *RAE1 homologue 1* (*RAH1*), which can also directly interact with STOP1 and promote its ubiquitination and degradation. As expected, double mutant *rah1rae1* shows reduced plant growth, while introduction of the *stop1* mutation (i.e., *rah1rae1stop1*) rescued plant growth [76].

STOP1-THO/TREX complex. Recent research has established the involvement of the THO/TREX complex in the regulation of nucleocytoplasmic STOP1 mRNA export, thus suggesting its regulation at a post-transcriptional level [77]. THO/TREX complex is a conserved eukaryotic multi-protein complex regulating transcription, mRNA processing, and export [78]. Thus, a mutant of *hyperrecombination protein 1* (*hpr1*), a subunit of the THO/TREX complex, demonstrated reduced expression of STOP1-regulated genes and the associated Al resistance. STOP1 mRNA was retained in the nucleus, which further resulted in decreased STOP1 protein abundance. Interestingly, the introduction of the *rae1* mutation to the *hpr1* background (i.e., *hpr1rae1* double mutant) partially rescued the reduced Al resistance and limited phosphate (_l_Pi) response in single *hpr1* mutants [77]. Similar biological functions have also been described for TEX1—another subunit of the THO/TREX complex. Mutant *tex1* plants exhibited reduced expression of STOP1-regulated genes, including *AtALMT1*. Unlike *hpr1*, *tex1* plants had no STOP1 mRNA accumulation in the nucleus; however, these plants had reduced levels of STOP1 protein. Similarly, double mutants (*tex1rae1*) showed partially rescued Al resistance and _l_Pi response phenotypes in comparison to single *tex1* plants [79]. In total, these results demonstrated that HPR1 and TEX1 have overlapping but distinct roles in the modulation of Al resistance and _l_Pi response through regulation of the expression of STOP1 downstream genes.

STOP1 SUMOylation. SUMOylation (SMALL UBIQUITIN-LIKE MODIFIER) is the process of reversible conjunction of SUMO to target proteins to modulate their function, stability, localisation, and/or activity [80]. Specifically, ELS-type SUMO protease 4 (AtESD4) was demonstrated to interact and deSUMOylate STOP1. *esd4* plants had increased level of STOP1 SUMOylation, which caused its increased association with the AtALMT1 promoter and, subsequently, enhanced *AtALMT1* expression. Three sites for SUMOylation were identified (K40, K212, and K395), and inhibition of STOP1 SUMOylation reduced STOP1 stability and the expression of STOP1 downstream genes [81]. Similarly, E3 SUMO ubiquitin ligase (AtSIZ1) is one of the proteins responsible for STOP1 SUMOylation. *siz1* plants showed reduced levels of both STOP1 SUMOylation and STOP1 protein, while the expression of STOP1-regulated genes (including *ALMT1*) was increased. Interestingly, *siz1* plants demonstrated increased Al stress resistance and _l_Pi response, which suggests STOP1-independent regulation of AtALMT1 expression [82].

### 5.6. The Role of Jasmonic Acid in ALMT Regulation

Jasmonic acid (JA) is another player involved in the regulation of Al stress-induced root growth inhibition. The expression of the major JA receptor *CORONATINE-INSENSITIVE1* (*COI1*) and the signalling regulator *bHLH-III-type transcription factor* (*MYC2*) was up-regulated in response to Al stress in the root tips. Accordingly, *coi1* and *myc2* plants showed reduced root growth inhibition in response to Al stress, while exogenous JA (in the form of methyl jasmonate) enhanced the Al-induced inhibition of root growth in WT plants. Interestingly, WT and *coi1* plants reacted to the treatment with other toxins (La^3+^, Cd^2+^, Cu^2+^, and Na^+^) in the same way. Furthermore, the expression level of *ALMT1* was two times higher in the roots of coi1 plants, while the expression of other related genes (*STOP1*, *MATE*, and *ALS1* and *3*) was not changed. Subsequently, malate exudation was higher in coi1 plants after Al treatment. However, *ALMT1* expression and the level of malate exudation after Al stress were not affected in the *myc2* plants. These data provide an additional JA-mediated signalling pathway regulating root growth inhibition in response to Al stress [83].

Recently, these results have been further elaborated in experiments on tomato (*Solanum lycopersicum* L.) where the crosstalk between JA and Al stress was analysed. Al stress up-regulated genes related to JA biosynthesis and signalling (such as *OXOPHYTODIENOIC ACID REDUCTASE 3*, *ALLENE OXIDE CYCLASE, COI1*, and *MYC2*), which further enhanced Al-induced root growth inhibition. Interestingly, both Al stress and JA treatment up-regulated several *WRKY* TFs (*3, 6, 16, 37, 39*, and *71*) and *ALMTs* (*3, 6*, and *7*), which were further analysed in *MYC2*-silenced and *JASMONIC ACID-INSENSITIVE 1* (*jai1*) plants. The obtained results suggest that Al stress and JA cross-talk in root growth inhibition is mediated via *SlALMT3* and six *SlWRKY* TFs as its upstream regulators [84].

In total, these data suggest a very complex and sophisticated network regulating *STOP1* and *ALMTs* expression and, subsequently, Al stress resistance and _l_Pi response. This network includes various TFs, hormones (such as GABA, JA, and ABA), and post-translational modifications (phosphorylation, SUMOylation, and ubiquitination) of the regulatory proteins. Despite the recent progress in our understanding of STOP1/ALMT regulation, further research in this direction may help to produce plants that are more tolerant to drought stress, acid, and Al-contaminated and low-phosphorus soils and more productive under adverse environmental conditions.

## 6. Conclusions and Future Prospects

Recent studies confirm the complex regulation of the ALMT family of proteins. In addition to the STOP1 TF, the expression and functioning of ALMTs are also regulated by plant hormones (primarily by GABA, but also JA and possibly others), ROS, metals (such as Al, Fe, and probably others), phosphorus availability in soil, and various post-translational modifications. Furthermore, the expression of *ALMT* genes affects plant immunity, proton tolerance, root development, and tolerance to abiotic stress factors. Further studies should characterise the molecular mechanisms underlying the crosstalk between these ALMT-regulating components.

Based on the discussed literature and the bioinformatic and evolutionary analysis conducted, we can formulate several directions for future investigation:The functional role of ALMT11 is still unknown. While it misses most conserved residues required for Al/malate/GABA recognition and transport, the presented sequence similarity (especially in the TM1 region) may be sufficient to form heterodimers with other ALMTs and, thus, deactivate them to act as a negative regulator.Currently, the role of ALMT14 isoforms 2 and 3 is unknown. Therefore, any research investigating the functions of these proteins would be beneficial, particularly their ability to recognise/transport Al^3+^ ions and/or malate and the ability to form homo/hetero dimers with other ALMTs. Recent results on full-length and truncated versions of rice (*Oryza sativa* L.) and wheat ALMTs suggest that they function as multimeric proteins, where combinations of ALMT subunits can affect channel function [85]. Similar experiments on ALMT11 and ALMT14 would greatly advance our understanding of ALMT functionality.The deletion in the CTD has been identified in several ALMTs. It would be interesting to identify how this deletion affects Al^3+^/malate recognition/transport function, or how it correlates with protein localisation and stability.The presence of the fusaric acid resistance protein-like (pfam13515) domain as the core of the TMD suggests that it may be the original form of the ALMT protein (before it acquired the CTD). So far, the ability of other Arabidopsis proteins possessing the fusaric acid resistance protein-like (pfam13515) domain to recognise/transport Al^3+^ or malate has not been studied.We have discussed several plant hormones interacting and regulating the STOP1/ALMT pathway of Al stress and _l_Pi response (GABA, JA, and ABA). However, analysis of the ALMT interactome suggests that other hormones may be involved. For example, ALMT1 was shown to interact with Arabidopsis histidine kinase 4, a cytokinin receptor [86], thus suggesting possible regulation by the cytokinin.Also, ALMT1 interacts with a cell wall-associated receptor-like protein kinase (WAK1) [87], which is known as a receptor of oligogalacturonides, is involved in wounding response, and is a regulator of cell wall synthesis.

## Figures and Tables

**Figure 1 plants-12-03167-f001:**
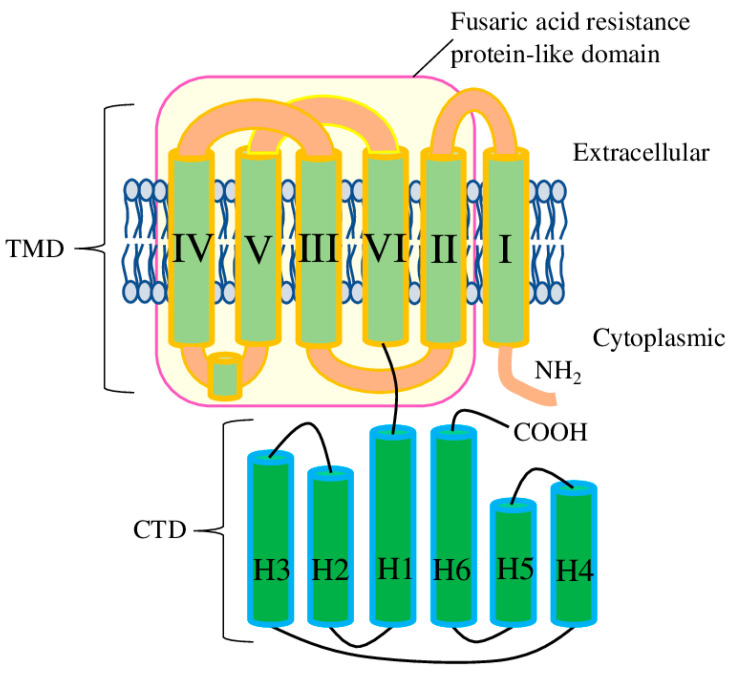
Schematic representation of the domain arrangement in one AtALMT1 subunit. The TMD (transmembrane domain) contains six TMs (transmembrane helices I to VI); the CTD (C-terminal cytosolic domain) contains six Hs (α-helix bundles 1 to 6); and the magenta rectangle represents the fusaric acid resistance protein-like (pfam13515) domain.

**Figure 2 plants-12-03167-f002:**
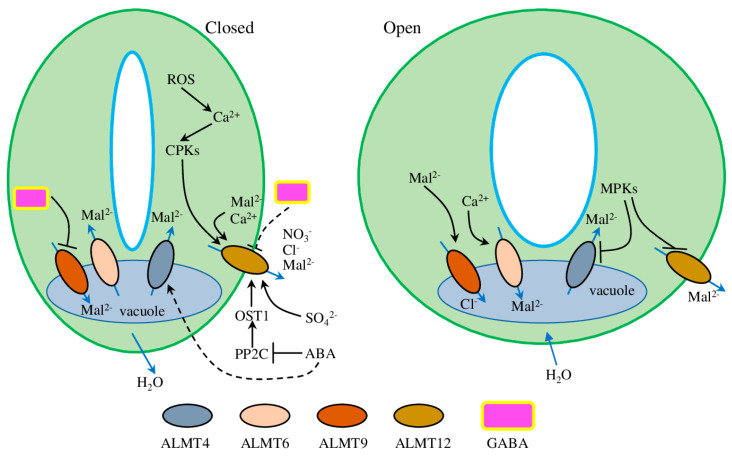
The role of ALMTs in stomatal opening and closure. ALMT proteins are responsible for the influx/efflux of compounds (blue arrows); black arrows represent activation and blunt arrows represent inhibition, while the dotted arrow represents indirect regulation. ROS—reactive oxygen species, Mal^2-^—malate, ABA—abscisic acid, PP2C—protein phosphatases type 2C, MPKs—mitogen-activated protein kinases, GABA—γ-aminobutyric acid, CPKs—calcium-dependent kinases.

**Figure 3 plants-12-03167-f003:**
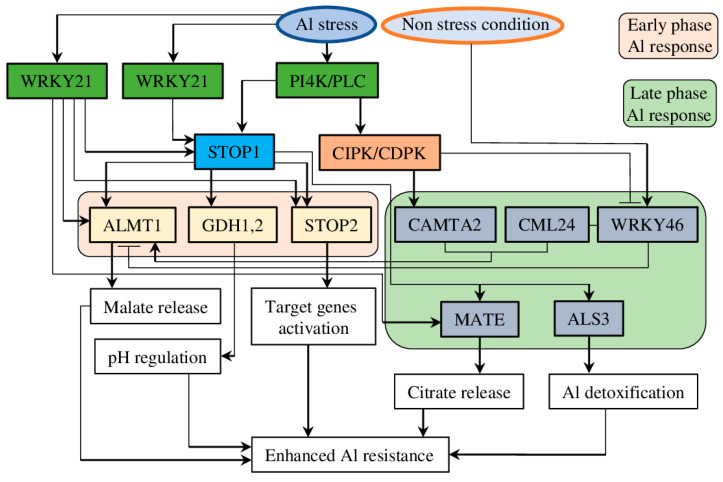
Schematic representation of STOP-mediated transcriptional regulation of target genes under Al stress and normal conditions. Genes responsible for early-phase Al response are depicted in a pink rectangle, and late-phase Al response genes are shown within a pale green rectangle. Regulators of STOP1 are depicted in a dark green rectangle. Black arrows represent activation, and blunt arrows represent inhibition.

**Figure 4 plants-12-03167-f004:**
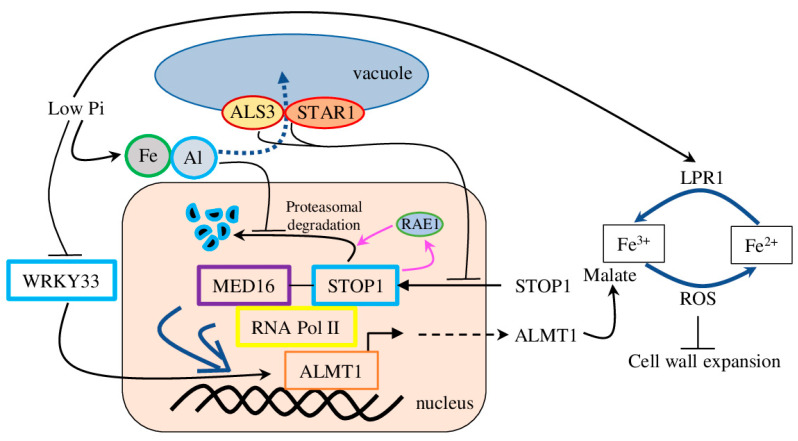
Schematic model for the role of *ALMT1* expression under low-Pi conditions. Under low-Pi and acidic conditions, Fe and Al accumulate in the cytoplasm, where they decrease the proteasomal degradation of STOP1, thereby promoting *ALMT1* transcription. The tonoplast-located ALS2–STAR1 proteins transport Fe/Al to the vacuole, thus reducing their concentration in the cytosol and, subsequently, reducing STOP1 accumulation in the nucleus and *ALMT1* transcription. ALMT1 exuded malate where it coupled with Fe and LOW PHOSPHATE ROOT 1 (LPR1); they generate ROS which inhibit cell wall expansion. Also, low Pi induced *LPR1/2* expression, which converts Fe^2+^ to Fe^3+^. MEDIATOR 16 (MED16) interacted with STOP1 and linked with RNA polymerase II (RNA Pol II) to promote the expression of *ALMT1* and malate efflux. The F-box protein Regulation of AtALMT1 Expression 1 (RAE1) interacted with STOP1 and promoted STOP1 ubiquitination and its further proteasomal degradation. At the same time, STOP1 promoted *RAE1* transcription, thus creating a negative feedback loop between RAE1 and STOP1 (depicted in magenta arrows). Black arrows represent activation, blunt black arrows represent repression, dashed arrows represent transfer between compartments, and blue arrows indicate that the reaction involved other proteins.

## Data Availability

Used sequences may be provided upon request.

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
