# Peer review of "Recent Updates on ALMT Transporters’ Physiology, Regulation, and Molecular Evolution in Plants"

_plants, 2023, doi:10.3390/plants12173167_

Round 1
Reviewer 1 Report
This manuscript aimed to summarize the current knowledge of ALMT family and assess their involvement in diverse physiological processes and comprehensive regulation mechanisms at the level of transcriptional regulation. In addition, the bioinformatic and phylogenetic analysis was accordingly conducted to provide new insights of the ALMT family proteins. The data for most part is well presented and interpreted. However, there are a few issues required to be addressed to be fully justified in the manuscript.
1. Add the recent research progress of ALMT family in the manuscript.
2. The progress of the ALMT family was mainly summarized and analyzed based on the research results of Arabidopsis. Can the regulatory mechanism of ALMT genes be explained in conjunction with major crops such as rice, wheat, corn, and soybean?
3. Check the correct writing of terms, molecular formula, genes, proteins, ions, etc., such as Al3+ in page 11.
Minor editing of English language required.
Author Response
Dear Editor and Reviewers,
We greatly appreciate your critical evaluation of our manuscript and helpful comments. Our reply to your comments would be provided point by point, where “A” stands for “Authors”, and “L” for “Lines”, where changes have been implemented. The language of the entire manuscript has been checked and corrected.
____________________________________________________________________________
This manuscript aimed to summarize the current knowledge of ALMT family and assess their involvement in diverse physiological processes and comprehensive regulation mechanisms at the level of transcriptional regulation. In addition, the bioinformatic and phylogenetic analysis was accordingly conducted to provide new insights into the ALMT family proteins. The data for the most part is well presented and interpreted. However, there are a few issues required to be addressed to be fully justified in the manuscript.
- Add the recent research progress of ALMT family in the manuscript.
A: Recent papers were added. The number of recent papers (from 2018 till now is 45 out of 93). - The progress of the ALMT family was mainly summarized and analyzed based on the research results of Arabidopsis. Can the regulatory mechanism of ALMT genes be explained in conjunction with major crops such as rice, wheat, corn, and soybean?
A: Indeed, Arabidopsis is the main model plant, the majority of experiments in plant science are conducted on Arabidopsis. Please, see cited papers with other species involved:
Lupinus albus [7], Vigna umbellate [8], Amaranthus mangostanus [9], Guzmania monostachia [11], Triticum aestivum [12], Brachypodium distachyon [24], Glycine max [45], [48], Nicotiana benthamiana and Oryza sativa [65][86], Malus domestica [68], Solanum lycopersicum [72]. - Check the correct writing of terms, molecular formula, genes, proteins, ions, etc., such as Al3+ in page 11.
A: the entire manuscript was checked and corrected
Reviewer 2 Report
ALMTs also play important roles in rice and maize that are staple crop worldwide. But in your manuscript, no advances related to rice and maize ALMTs were presented.
Some important references were not cited in your manuscript, for instance, (1) Structural basis of ALMT1-mediated aluminum resistance in Arabidopsis. Cell Res,2022,32(1):89-98.
(2) The transcription factor BZR1 represses ALMT1 transcription via competing with STOP1 (please see Liu T, Deng S, Zhang C, Yang X, Shi L, Xu F, Wang S, Wang C. Brassinosteroid signaling regulates phosphate starvation-induced malate secretion in plants. J Integr Plant Biol. 202365(5):1099-1112.
In this paper, the authors revealed that BZR1 represses the expression of ALMT1 via competing with STOP1.
(3) In the section of "Conclusion and future prospective", you just discuss the roles of ALMT in Arabidopsis. Thus you had better have pointed out that the functions of Arabidopsis ALMT11, ALMT14 isoforms are unknown, to avoid misleading readers.
Some minor concerns:
(1) in page 1, "abscisic acid" should be "abscisic acid (ABA)".
(2) what is 1Pi?? Please check it in whole manuscript.
--------------------
Extra Comments for the authors' consideration:
(1) I would like to see the introduction followed by the section of bioinformatic analysis of ALMT family proteins and that of molecular evolution of ALMTs, and then the section of ALMTs in the regulation of Stomata/Guard Cell.
(2) As to the section of "4.4. ALMT phosphorylation", as the author reviewed the regulation of ALMTs in transcription level, hence they should have presented the regulation of ALMTs in protein level such as phosphorylation in parellel. Accordingly, it would be better that the authors firstly review the transcriptional regulation of ALMTs, and then the translational regulation of ALMTs.
Please pay attention to the English mistakes:
(1) "cis" should be italic in the whole manuscript.
(2) in page 2, "then it was thought before" should be "than it was thought before".
Author Response
Dear Editor and Reviewers,
We greatly appreciate your critical evaluation of our manuscript and helpful comments. Our reply to your comments would be provided point by point, where “A” stands for “Authors”, and “L” for “Lines”, where changes have been implemented. The language of the entire manuscript has been checked and corrected.
____________________________________________________________________________
ALMTs also play important roles in rice and maize that are staple crop worldwide. But in your manuscript, no advances related to rice and maize ALMTs were presented.
A: recent research on other species (including rice and maize) was added.
There are two publications dedicated to maize ALMTs from 2010 and 2012 (10.1371/journal.pone.0009958 and 10.1111/j.1365-3040.2011.02479.x), which, however, were not included in our review because they did not fall into the category “recent”.
Some important references were not cited in your manuscript, for instance, (1) Structural basis of ALMT1-mediated aluminum resistance in Arabidopsis. Cell Res,2022,32(1):89-98.
A: This paper was cited (please, see ref 68 in the original manuscript).
(2) The transcription factor BZR1 represses ALMT1 transcription via competing with STOP1 (please see Liu T, Deng S, Zhang C, Yang X, Shi L, Xu F, Wang S, Wang C. Brassinosteroid signalling regulates phosphate starvation-induced malate secretion in plants. J Integr Plant Biol. 202365(5):1099-1112.
In this paper, the authors revealed that BZR1 represses the expression of ALMT1 via competing with STOP1.
A: Suggested reference was added, please, see 3rd paragraph of section 3.4.
(3) In the section of "Conclusion and future prospective", you just discuss the roles of ALMT in Arabidopsis. Thus you had better have pointed out that the functions of Arabidopsis ALMT11, ALMT14 isoforms are unknown, to avoid misleading readers.
A: Thank you for your note. We have clearly indicated that the functions of ALMT11 are unknown, please, see 5th and 6th sentences of section 4 and the very first sentence of the 1st bullet of section 6.
and ALMT14: 7th and 8th sentences of section 4 and the very first sentence of the 2nd bullet of section 6.
Some minor concerns:
(1) in page 1, "abscisic acid" should be "abscisic acid (ABA)".
A: corrected as suggested
(2) what is 1Pi?? Please check it in whole manuscript.
A: lPi stands for “limited phosphate”, the full form for this abbreviation was additionally added on page 6
--------------------
Extra Comments for the authors' consideration:
(1) I would like to see the introduction followed by the section of bioinformatic analysis of ALMT family proteins and that of molecular evolution of ALMTs, and then the section of ALMTs in the regulation of Stomata/Guard Cell.
A: The sections were shifted as suggested.
(2) As to the section of "4.4. ALMT phosphorylation", as the author reviewed the regulation of ALMTs in transcription level, hence they should have presented the regulation of ALMTs in protein level such as phosphorylation in parellel. Accordingly, it would be better that the authors firstly review the transcriptional regulation of ALMTs, and then the translational regulation of ALMTs.
A: Section 4 has been already shifted and modified. The translational regulation section was also shifted, please, see section 3.
Comments on the Quality of English Language
Please pay attention to the English mistakes:
(1) "cis" should be italic in the whole manuscript.
A: “cis” corrected to italic throughout the entire manuscript. However, we could not guarantee that it will not be removed by editors
(2) in page 2, "then it was thought before" should be "than it was thought before".
A: corrected as suggested
Reviewer 3 Report
The Recent Updates on ALMT Transporter's Physiology, Regulation and Molecular Evolution in….
The MS is based on an important aspect and describes some interesting facts. There are several feedbacks to improve the MS.
There are several mistakes related to English language and sentence formation, which makes many of them unclear doesn’t express the authors intentions. I can write only few, not possible for whole MS
For example
Abstract 1st line: are a membrane protein family
Abstract 2nd line: tolerance to environmental Al3+
Page 3:1st para: described with Cl- -dependent defective sto
Page 4 2nd para: Because ALMT proteins have a GABA-binding motif
Sentence structure should be corrected by authenticated English assistance.
Supplementary file is not openable. So data claimed there is in question
There is very much less mention of Al toxicity, where as one is reading aluminum for understanding toxicity issues and to overcome strategies. Address strongly the toxicity.
Abstract 1st sentence should start with importance of aluminum and toxicity and then come to ALMT.
Section 5.1: there is no data shown for the text described there
Supplementary 2 is again not openable
Section 6: prospective spelling wrong
Major data is remaining in supplementary files. So review comments are also not complete.
Newest references should be discussed and added from last 5 years
The Recent Updates on ALMT Transporter's Physiology, Regu[1]lation and Molecular Evolution in….
The MS is based on an important aspect and describes some interesting facts. There are several feedbacks to improve the MS.
There are several mistakes related to English language and sentence formation, which makes many of the them unclear doesn’t express the authors intentions. I can write only few not possible for whole MS
For example
Abstract 1st line: are a membrane protein family
Abstract 2nd line: tolerance to environmental Al3+
Page 3:1st para: described with Cl- -dependent defective sto
Page 4 2nd para: Because ALMT proteins have a GABA-binding motif
Sentence structure should be corrected by authenticated English assistance.
Supplementary file is not openable. So data claimed there is in question
There is very much less mention of Al toxicity, where as one is reading aluminum for understanding toxicity issues and to overcome strategies. Adress strongly the toxicity.
Abstract 1st sentence should start with importance of aluminum and toxicity and then come to ALMT.
Section 5.1: there is no data shown for the text described there
Supplementary 2 is again not openable
Section 6: prospective spelling wrong
Major data is remaining in supplementary files. So review comments are also not complete.
Newest references should be discussed and added from last 5 years
Author Response
Dear Editor and Reviewers,
We greatly appreciate your critical evaluation of our manuscript and helpful comments. Our reply to your comments would be provided point by point, where “A” stands for “Authors”, and “L” for “Lines”, where changes have been implemented. The language of the entire manuscript has been checked and corrected.
____________________________________________________________________________
The Recent Updates on ALMT Transporter's Physiology, Regulation and Molecular Evolution in….
The MS is based on an important aspect and describes some interesting facts. There are several feedbacks to improve the MS.
There are several mistakes related to English language and sentence formation, which makes many of them unclear doesn’t express the authors intentions. I can write only few, not possible for whole MS
For example
Abstract 1st line: are a membrane protein family
A: the sentence was corrected
Abstract 2nd line: tolerance to environmental Al3+
A: the sentence was corrected
Page 3:1st para: described with Cl- -dependent defective sto
A: the sentence was corrected
Page 4 2nd para: Because ALMT proteins have a GABA-binding motif
A: the sentence was corrected
Sentence structure should be corrected by authenticated English assistance.
A: the entire manuscript was edited
Supplementary file is not openable. So data claimed there is in question
A: We have checked the supplementary file. It works, This comment will be re-directed to the Editors
There is very much less mention of Al toxicity, where as one is reading aluminum for understanding toxicity issues and to overcome strategies. Address strongly the toxicity.
Abstract 1st sentence should start with importance of aluminum and toxicity and then come to ALMT.
A: more recent research dedicated to Al toxicity was added. The abstract was modified as suggested.
Section 5.1: there is no data shown for the text described there
A: There is no data file, associated with this section. One may easily download a full set of ALMT-related sequences from the mentioned databases. However, their cleaning, checking and verification have required significant efforts (due to their high number). Therefore, we could not make it freely available, but would willingly share it for a specifically designated purpose.
Supplementary 2 is again not openable
A: We are sorry for this inconvenience. This comment will be re-directed to the Editors.
Section 6: prospective spelling wrong
A: “prospective” was corrected to “prospects”
Major data is remaining in supplementary files. So review comments are also not complete.
A: We are sorry for this inconvenience. This comment will be re-directed to the Editors.
Newest references should be discussed and added from last 5 years
A: more recent research has been added. In the current version there 45 recent references (2018 till now).
Round 2
Reviewer 2 Report
You have improved your manuscript and answer most of my concerns. But I still have two minor concerns:
1. In page 14, "the functional role of ALMT11 is still unknown" had better change to "the functional role of Arabidopsis ALMT11 is still unknown".
2. In page 14, "Currently, the role of ALMT14 isoforms 2 and 3 is unknown" had better change to "Currently, the role of Arabidopsis ALMT14 isoforms 2 and 3 is unknown".